# Free Fatty Acids’ Level and Nutrition in Critically Ill Patients and Association with Outcomes: A Prospective Sub-Study of PermiT Trial

**DOI:** 10.3390/nu11020384

**Published:** 2019-02-13

**Authors:** Yaseen M. Arabi, Waleed Tamimi, Gwynne Jones, Dunia Jawdat, Hani Tamim, Hasan M. Al-Dorzi, Musharaf Sadat, Lara Afesh, Maram Sakhija, Abdulaziz Al-Dawood

**Affiliations:** 1College of Medicine, King Saud bin Abdulaziz University for Health Sciences, King Abdullah International Medical Research Center, Riyadh 11426, Saudi Arabia; TamimiW@NGHA.MED.SA (W.T.); aldorziha@NGHA.MED.SA (H.M.A.-D.); sadatmu@ngha.med.sa (M.S.); SakkijhaM@NGHA.MED.SA (M.S.); dawooda@ngha.med.sa (A.A.-D.); 2Intensive Care Department, King Abdulaziz Medical City, Riyadh 11426, Saudi Arabia; 3Department of Clinical Laboratory, King Abdulaziz Medical City, Riyadh 11426, Saudi Arabia; 4Department of Medicine, Division of Critical Care Medicine, University of Ottawa, Ottawa Hospital Research Institute, Ottawa KIH 8L6, Canada; GJones@ottawahospital.on.ca; 5College of Medicine, King Saud bin Abdulaziz University for Health Sciences, Cord Blood Bank, King Abdullah International Medical Research Center, Riyadh 11426, Saudi Arabia; JawdatD@NGHA.MED.SA; 6Department of Internal Medicine, American University of Beirut—Medical Center, Beirut 110-236, Lebanon; hani_t@hotmail.com; 7King Abdullah International Medical Research Center, Riyadh 11426, Saudi Arabia; Afeshla@ngha.med.sa

**Keywords:** critical illness, nonesterified fatty acids, caloric restriction, insulin resistance, metabolic syndrome

## Abstract

Objectives: The objectives of this study were to evaluate the clinical and nutritional correlates of high free fatty acids (FFAs) level in critically ill patients and the association with outcomes, and to study the effect of short-term caloric restriction (permissive underfeeding) on FFAs level during critical illness. Patients/Method: In this pre-planned sub-study of the PermiT (Permissive Underfeeding vs. Target Enteral Feeding in Adult Critically Ill Patients) trial, we included critically ill patients who were expected to stay for ≥14 days in the intensive care unit. We measured FFAs level on day 1, 3, 5, 7, and 14 of enrollment. Of 70 enrolled patients, 23 (32.8%) patients had high FFAs level (baseline FFAs level >0.45 mmol/L in females and >0.6 mmol/L in males). Results: Patients with high FFAs level were significantly older and more likely to be females and diabetics and they had lower ratio of partial pressure of oxygen to the fraction of inspired oxygen, higher creatinine, and higher total cholesterol levels than those with normal FFAs level. During the study period, patients with high FFAs level had higher blood glucose and required more insulin. On multivariable logistic regression analysis, the predictors of high baseline FFAs level were diabetes (adjusted odds ratio (aOR): 5.36; 95% confidence interval (CI): 1.56, 18.43, *p* = 0.008) and baseline cholesterol level (aOR, 4.29; 95% CI: 11.64, 11.19, *p* = 0.003). Serial levels of FFAs did not differ with time between permissive underfeeding and standard feeding groups. FFAs level was not associated with 90-day mortality (aOR: 0.49; 95% CI: 0.09, 2.60, *p* = 0.40). Conclusion: We conclude that high FFAs level in critically ill patients is associated with features of metabolic syndrome and is not affected by short-term permissive underfeeding.

## 1. Introduction

Fatty acids are a major source of fuel in the body and play an important role in cell signaling [1,2]. Free fatty acids (FFAs) are nonesterified fatty acids that are released by the hydrolysis of triglycerides (triglyceride molecule is composed of three fatty acid molecules bound to glycerol) within the adipose tissue by lipoprotein lipase. They circulate in the blood protein-bound, serving as an energy source for tissues [1,3]. Chronically elevated FFAs level has been observed in obese people and in diabetic patients and is associated with insulin resistance and with sudden death in middle-aged men without known ischemic heart disease [1,3,4]. 

Acutely, FFAs level is often increased during critical illness and may contribute to organ dysfunction. Critical illness is characterized by hypercatabolic state and by a change in the contribution of the endogenous protein, fat, and carbohydrate sources to oxidative fuel [5]. Lipolysis is accelerated by the high catecholamine and other stress hormone milieu leading to increased release of FFAs from adipocytes, thus, increasing FFAs level [6]. Insulin resistance during critical illness impairs the use of FFAs for energy, and, thus, contributes to increased FFAs level [7]. Heparin given during critical illness may also increase FFAs level by activating lipoprotein lipase [8]. FFAs may have toxic effects by increasing reactive oxygen species leading to cell death and necrosis [9] and by depressing the immune cell function [10]. In addition, FFAs potentiate insulin resistance and impair glucose metabolism by inhibiting glucose oxidation and by stimulating protein kinase C [1,10]. In an acute setting, elevated FFAs level has been associated with the development of acute lung injury in at-risk patients with sepsis, trauma, and pancreatitis and after on-pump coronary artery bypass grafting [8,11,12].

Can FFAs level in critically ill patients be modulated by short-term caloric restriction (permissive underfeeding)? Normally, serum FFAs level increases during fasting and exercise and after a fatty meal. FFAs level goes down postprandially due to the anti-lipolytic effect of insulin that is released after carbohydrate intake [13]. On the other hand, caloric restriction and weight loss lead to a lowering of FFAs level and can attenuate FFAs-induced hepatic insulin resistance in obese healthy patients [14,15,16]. However, the effect of caloric restriction on serum FFAs level has not been investigated in critically ill patients. The aims of this study were (1) to evaluate the clinical and nutritional correlates of high FFAs level in critically ill patients and the association with outcomes, and (2) study the effect of short-term caloric restriction (permissive underfeeding) on FFAs level during critical illness. 

## 2. Materials and Methods

### 2.1. Patients

This is a pre-planned sub-study of the PermiT [17] (Permissive Underfeeding vs. Target Enteral Feeding in Adult Critically Ill Patients—ISRCTN68144998) trial, in which critically ill patients were randomized to permissive underfeeding (40–60% of calculated caloric requirements) or standard feeding (70–100%) for up to 14 days while maintaining similar protein intake in both groups. The trial found no difference in the primary endpoint of 90-day mortality. In this sub-study which was separately funded by King Abdulaziz City for Science and Technology (KACST), Riyadh, Saudi Arabia (Grant Number—AT 32-25 KACST), we enrolled consecutive patients from the PermiT trial at King Abdulaziz Medical City, Riyadh, Saudi Arabia between September 2012 and September 2014 who were expected to stay ≥14 days in the intensive care unit as judged by their primary team. A separate informed consent was obtained for participation in this sub-study. The study was approved by the Institutional Board Review of the Ministry of the National Guard Health Affairs, Riyadh, Saudi Arabia. Blood was collected at the time of enrollment (baseline or study day 1) within 48 h of ICU admission and on days 3, 5, 7, and 14. Serum was prepared from the blood samples by centrifugation at 4 °C at 1600 g for 20 min and divided into aliquots. These aliquots were stored immediately in a designated freezing area at −70 °C to be analyzed once the sample size was completed. The samples were analyzed blindly, and then the sample codes were broken. 

### 2.2. Free Fatty Acids Measurement

The measurement of FFAs was performed in Bioscientia reference laboratory in Germany using an in-vitro enzymatic calorimetric assay with Wako-NEFA-HR (2) reagent (Wako-chemicals, Neuss, Germany) [18]. In this method, FFAs with the coexistence coenzyme A (CoA) and adenosine-5’-triphosphate (ATP) disodium salt were converted to Acyl-CoA, adenosine monophosphate (AMP) and pyrophosphoric acid by the action of Acyl-CoA synthetase (ACS). The Acyl-CoA was oxidized yielding 2,3-trans-enoyl-CoA and hydrogen peroxide (H_2_O_2_) by the action of acyl-CoA oxidase. In the presence of peroxidase, H_2_O_2_ yielded a blue–purple pigment by quantitative oxidation condensation with 3-Methyl-N-Ethyl-N-(β-Hydroxyethyl)-Aniline (MEHA) and 4-aminoantipyrine (4AA). FFAs level was obtained by measuring absorbance at the blue and purple color at wavelengths of 546 nm and 660 nm.

The normal fasting serum FFAs level is 0.1 to 0.45 mmol/L for females and 0.1 to 0.6 mmol/L for males. However, FFAs level in the critically ill is poorly studied. In the current study, patients with FFAs more than 0.45 mmol/L in females and 0.6 mmol/L in males were considered to have high FFAs level; otherwise FFAs level was considered normal.

### 2.3. Data Collection

Baseline data included demographics, Acute Physiology and Chronic Health Evaluation Scores (APACHE) II [19], presence of sepsis upon admission, Sequential Organ Failure Assessment (SOFA) score [20], the ratio of partial pressure of arterial oxygen to the fraction of inspired oxygen (PaO_2_:FiO_2_), Glasgow coma scale and various laboratory results (baseline blood glucose, hemoglobin, international normalized ratio(INR), platelets, bilirubin, creatinine, C-reactive protein, albumin, pre-albumin, transferrin, 24-h urinary urea nitrogen excretion, and nitrogen balance). For the intervention period, which lasted for up to 14 days, we collected daily nutritional data (feeding formula and calories from enteral feeds, propofol, intravenous dextrose, and parenteral nutrition), insulin dose for hyperglycemia management, daily blood glucose, and use of certain medications, such as aspirin, beta-blockers and statins. We noted the daily carbohydrate, fat, and protein calories from enteral and parenteral sources and then calculated the total fat-to-carbohydrate ratio by dividing fat calories by carbohydrate calories. 

The outcomes evaluated in this study were 28-, 90-, and 180-day all-cause mortality. Other outcomes included hospital and ICU mortality, incident renal replacement therapy, ICU-associated infections [21], ICU and hospital length of stay (LOS), and mechanical ventilation duration. In addition, ICU-free days, renal replacement therapy-free days, and ventilator-free days were also calculated. 

### 2.4. Statistical Analysis

We reported categorical variables as frequencies with percentages and continuous variables as medians with quartile 1 and 3 (Q1, Q3). We compared categorical variables using chi-square or Fisher’s exact test and continuous variables using Mann–Whitney U test. We examined Pearson correlation among the following baseline variables FFAs level, age, body mass index, total cholesterol, high-density lipoprotein (HDL) cholesterol, low-density lipoprotein (LDL) cholesterol, non-HDL cholesterol, triglycerides, glucose, and hemoglobin A1c. In addition, multivariable logistic regression analysis was performed to assess the predictors of high FFAs level. We entered in the model a priori decided baseline variables that were of clinical interest and/or had significant association with high FFAs level by univariable analysis (*p* ≤ 0.05) which included age, gender, body mass index (BMI), APACHE II, diabetes, triglycerides, LDL cholesterol, HDL cholesterol, medical admissions (vs. non-medical admissions), and randomization (permissive vs. standard feeding). We also carried out a linear mixed model to test whether FFAs level is affected over time with permissive underfeeding compared to standard feeding. We carried out logistic and linear regression models to examine the association between FFAs level and outcomes adjusting for age, gender, BMI, APACHE II, diabetes, triglycerides, LDL cholesterol, HDL cholesterol, non-HDL cholesterol, and medical admissions (vs. non-medical admissions). A two-tailed *p* value < 0.05 was considered statistically significant. The results were expressed as adjusted odds ratio (aOR) or parameter estimate with 95% confidence intervals (95%CI). All statistical analyses were performed using SAS version 9.2 (SAS Institute, Cary, NC, USA).

## 3. Results

### 3.1. Characteristics of Patients

Of the 70 patients included in the study (Appendix A), 23 (32.8%) had high FFAs level (median 0.74 mmol/L (Q1, Q3: 0.63, 1.06) and 47 (67.1%) had normal FFAs level (0.34 mmol/L (0.22, 0.45))). Patients with high FFAs level were significantly older, more likely to be females and diabetic, had higher HgbA1c, creatinine and non-HDL and LDL cholesterol levels, and had lower PaO_2_:FiO_2_ ratio compared with patients with normal FFAs (Table 1). There were significant correlations between FFAs level and total cholesterol (*r* = 0.45, *p* = 0.0001), non-HDL cholesterol (r = 0.38, *p* = 0.002), HDL cholesterol (*r* = 0.30, *p* = 0.01) and LDL cholesterol (*r* = 0.43, *p* = 0.0003), and age (*r* = 0.40, *p* = 0.0006) (Appendix A).

Table 2 shows the nutritional data during the study period and the trial co-interventions. The total daily caloric intake was 1065.6 kcal (Q1, Q3: 785.0, 1496.1) in patients with high FFAs level and 1071.0 kcal (Q1, Q3: 810.0, 1368.5) in patients with normal FFAs level (*p* = 0.73). The median calories from propofol were only 17.3 kcal (Q1, Q3: 0.0, 102.6) in patients with high FFAs level and 70.3 (Q1, Q3: 19.1, 134.2) in patients with normal FFAs level (*p* = 0.06). This corresponded to the following doses of propofol; 1.2 mcg/kg/min (Q1, Q3: 0.0, 9.0) in patients with high FFAs level and 5.2 mcg/kg/min (Q1, Q3: 1.4, 11.2) in patients with normal FFAs level (*p* = 0.06). The daily protein intake was 55.4 g (Q1, Q3: 44.2, 66.4) in patients with high FFAs level and 61.9 g (Q1, Q3: 42.8, 77.1) in patients with normal FFAs level (*p* = 0.37). The baseline blood glucose was similar in the two groups; however, during the study period, the glucose level was significantly higher in patients with high FFAs level compared to patients with normal FFAs level (9.5 mmol/L (Q1, Q3: 7.6, 11.9) compared to 7.7 mmol/L (Q1, Q3: 6.5, 10.3) *p*= 0.05) with higher use of insulin (18.9 units per day (Q1, Q3: 3.8, 37.5) compared to 0.0 units per day (Q1, Q3: 0.0, 14.9) *p* = 0.003). Additionally, more patients in the high FFAs level received disease-specific formulae, renal replacement therapy, aspirin and statins during ICU stay.

### 3.2. Predictors of High FFA and FFA Levels During the Study Period

On multivariable logistic regression analysis, the independent predictors of high FFAs level were diabetes (aOR, 5.36; 95% CI, 1.56, 18.43; *p* = 0.008), and baseline cholesterol (aOR, 4.29; 95% CI, 1.64, 11.19; *p* = 0.003). Figure 1, panel A shows the serial levels of FFAs for patients with high FFAs and normal FFAs. Figure 1, panel B shows FFAs level in patients who received permissive underfeeding and standard feeding. The FFAs level was not different between the two feeding strategies. 

### 3.3. Outcomes of Patients

There was no significant difference in crude mortality between patients with high and normal FFAs level (Table 3). Incident of renal replacement therapy was more frequent in patients with high FFAs level (6/23 (27.3%) compared to 2/47 (4.7%), *p* = 0.009). Multiple variable analyses adjusting for age, gender, BMI, APACHE II, diabetes, triglycerides, LDL cholesterol, HDL cholesterol, non-HDL cholesterol, and medical admissions (vs. non-medical admissions) showed no significant association between high FFAs level and 90-day mortality (aOR 0.49, 95% CI 0.09, 2.60, *p* = 0.40) or any other study outcomes (Table 3). 

## 4. Discussion

In this study, we evaluated serum FFAs level in critically ill patients. We found that FFAs level was elevated at baseline in 32% of patients, and that it was associated with features of the metabolic syndrome. FFAs level was not affected by permissive underfeeding versus standard feeding. High FFAs level appear to be largely a reflection of the underlying metabolic condition of the patient rather than the critical illness itself.

Our study provides a characterization of critically ill patients with high FFAs level. We found that high FFAs level correlated highly with other lipid profile parameters (total, HDL and LDL cholesterol, but not triglycerides) and with age. Compared to those with normal FFAs level, patients with high FFAs level were at baseline significantly older, more likely to be diabetic, had higher HgbA1c, blood glucose, creatinine and non-HDL and LDL cholesterol concentrations, and had more hypoxemia (as reflected lower pO2: FiO2 ratio) despite lack of differences in APACHE II scores, SOFA scores, and vasopressor demands. During ICU stay, patients with high FFAs level had increased demand for insulin, disease-specific nutrition therapy, RRT, aspirin, and statins. The differences suggest the association of high FFAs level with metabolic syndrome. There was no difference in BMI, between the two groups; however, BMI is known to have its limitations in predicting obesity [22]. Because of these differences, we carried out multivariable analyses to account for the confounding effect of some of these variables on clinical outcomes. These analyses show that high FFAs level is not associated independently with clinical outcomes. Interestingly, patients with high FFAs level had less 24-h urinary nitrogen excretion and less negative nitrogen balance. This may be related to lower muscle mass in this older population and more frequent insulin therapy.

Normally, FFAs are elevated during fasting and exercise, and their level drops postprandially after carbohydrate-rich meals. FFAs level are elevated in obesity and diabetes [1,3]. In acute critical illness, where lipolysis increases, serum FFAs level increases [6]. Our study demonstrated that one-third of critically ill patients had high FFAs level; most (63.6%) of these patients were diabetics. We found that baseline cholesterol level and diabetes were independent risk factors for high FFAs level on multivariable logistic regression analysis. The effect of propofol on FFAs level is uncertain. In an experiment on dogs undergoing general anesthesia, high concentration of propofol (200 and 400 mcg/kg/min) were associated with increased FFAs level, although a study in humans undergoing general anesthesia for cardiopulmonary bypass showed that propofol (50 mcg/kg/min) compared to midazolam did not alter serum FFAs level [23,24]. In our study, categorization of patients into high and low FFAs level was based on baseline serum specimens, and doses of propofol preceding enrolment were not collected. Our study was not designed to specifically address the effect of propofol on FFAs level. Nevertheless, the doses of propofol that were used during the ICU stay were on average much lower than what was used in these studies, and, therefore, the effect of propofol on FFAs level in our cohort is likely to be small. The slightly lower dose of propofol given to patients with high-FFAs level was likely to be related to being older and more susceptible to sedation. Therefore, these patients would normally receive smaller doses of propofol.

In addition to being a fuel source, FFAs have multiple other physiologic effects. FFAs are associated with insulin resistance and impaired glucose metabolism by inhibiting glucose oxidation and by stimulating protein kinase C [1,10,25]. They may also stimulate the autophagy of pancreatic beta cells [26]. In our study, patients with high FFAs level had higher blood glucose and required more insulin therapy during the ICU stay even though the baseline blood glucose level was similar in patients with high and normal FFAs, suggesting an association of FFAs and insulin resistance in ICU patients.

FFAs may also affect the course of acute critical illness. FFAs were found to exacerbate hyperglycemia-induced Toll-like receptor expression and activity in monocytic cells, increase superoxide release, enhance Nuclear factor-κB activity, and induce the release of proinflammatory factors in diabetics [27]. Whether FFAs affect inflammation in critically ill patients is less clear. In a porcine endotoxemia model, infusing lipids at two different concentrations was associated with no differences in plasma tumor necrosis factor-α, interleukin6, and leucocytes between animals with low and high FFAs suggesting that FFAs does not play a significant pro-inflammatory mediator effect [28]. However, FFAs have been implicated in the pathogenesis of acute respiratory distress syndrome and has been identified as a prognostic factor for this syndrome. In a lipopolysaccharide-induced acute lung injury model, a 15-fold increase in free oleic acid was observed in bronchoalveolar lavage fluid from mice 8 h after lipopolysaccharide application [20]. The FFA, oleic acid has been demonstrated to be elevated in patients with ARDS (acute respiratory distress syndrome) and in patients at-risk for ARDS [11,29]. Patients with sepsis demonstrated a six-fold increase in plasma oleic acid levels compared to healthy volunteers [19]. In addition, FFAs are elevated in the blood of patients with sepsis who are at increased risk for ARDS [30]. The exact mechanism of FFAs-associated lung injury is unclear; however, FFAs have been shown to increase permeability and to impair transepithelial active sodium transport mechanisms in the lung, and could, thus, promote alveolar edema formation and prevent edema resolution [31]. In our study, where almost all patients were on mechanical ventilation at baseline, hypoxemia was more significant in patients with high FFAs compared with patients with normal FFAs (median PaO2: FiO2 ratio was 115 vs. 200, *p* = 0.02), a finding that may be in line with the association of FFAs and lung injury. Whether FFAs are toxic to the kidneys is unclear. In an animal study, FFAs led to severe tubulointerstitial damage [32]. FFAs and their metabolites have been implicated in renal cell injury and development of chronic kidney in patients with the metabolic syndrome [33]. In our study, patients with high FFAs had a higher rate of new renal replacement therapy, although this association became not significant in multivariable analysis. This finding suggests that the observed crude association may be related to other confounders that put these patients at a higher risk for acute kidney injury; for example, patients with high FFAs level were more likely to be diabetics and had higher baseline creatinine compared to patients with normal FFAs level.

Weight loss leads to a lowering of FFAs level in the long run and can attenuate FFAs-induced hepatic insulin resistance in obese healthy patients [14,15,16]. However, the effects of short-term caloric restriction are different. In one study, 11 subjects were fed for two periods of 6 days with hypo- and eucaloric diet with the same macronutrient composition in random order [34]. At 6 days, fasting FFAs significantly increased with the hypocaloric diet compared with the eucaloric diet [34]. Whether the macronutrient composition affects FFAs was investigated in an animal model, and the study found that energy-restricted high-fat versus low-fat diet did not result in different FFA levels [35]. In the current study, serial levels of FFAs did not differ with time between patients receiving permissive underfeeding and standard feeding. 

The study results should be interpreted taking into considerations its strengths and limitations. Strength include that data came from a randomized controlled trial, and that serial measurements of FFAs were obtained. The limitations include the sample size, which makes the study underpowered to detect a mortality difference. We measured total FFAs but not individual levels of each FFAs. In addition, the study included patients who had an expected duration of ICU stay ≥14 days and, thus, the results may not be generalizable to patients who have a shorter stay.

In conclusion, we found that serum FFAs level was elevated in almost one-third of critically ill patients. High FFAs level was associated with features of the metabolic syndrome and was not affected by short-term moderate caloric restriction.

## Figures and Tables

**Figure 1 nutrients-11-00384-f001:**
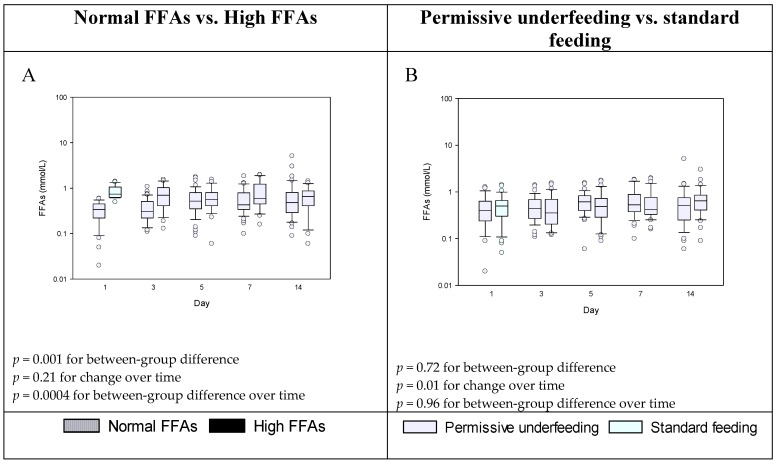
Serial levels of FFAs in patients with high and normal FFAs (Panel **A**) and in patients who received permissive underfeeding and standard feeding (Panel **B**). *p* values for between-group differences and between-group differences over time are provided using mixed linear model.

**Table 1 nutrients-11-00384-t001:** Baseline characteristics of patients with high and normal free fatty acids (FFAs) level.

Variable	High FFAs Level * *n* = 23	Normal FFAs Level * *n* = 47	*p*-Value
**Age** (years), median (Q1, Q3)	65 (30.9, 70.6)	30.2 (24.4, 61.1)	0.006
**Female sex**, *n* (%)	10 (45.5)	6 (13.0)	0.003
**Height** (cm), median (Q1, Q3)	164 (150, 168)	170 (165, 173)	0.002
**Weight** (kg), median (Q1, Q3)	80 (56, 93)	75 (60, 96)	0.84
**Body mass index** (kg/m^2^), median (Q1, Q3)	27.9 (25.3, 35.1)	27.3 (21.2, 32.5)	0.13
**Diabetes**, *n* (%)	14 (63.6)	11 (23.9)	0.002
**Admission category**, *n* (%)			0.26
Medical	17 (77.3)	19 (41.3)	
Surgical	0 (0.00)	4 (8.7)	
Post-operative trauma	5 (22.7)	23 (50.0)	
**Patients on renal replacement therapy**, *n* (%)	0 (0.00)	3 (6.5)	0.22
**APACHE II**, median (Q1, Q3)	21 (15, 27)	18 (14, 23)	0.17
**Mechanical ventilation**, *n* (%)	22 (100)	44 (96.7)	0.32
**Sepsis at admission**, *n* (%)	5 (22.7)	8 (17.4)	0.60
**SOFA Score Day 1**, median (Q1, Q3)	12 (8, 13)	10 (8, 12)	0.05
**PaO**_**2**_**: FiO**_**2**_**ratio**, median (Q1, Q3)	115 (73, 171)	200 (114, 311)	0.02
**Glasgow coma scale**, median (Q1, Q3)	4 (3, 6)	3 (3, 5)	0.11
**Vasopressor**, *n* (%)	14 (63.6)	29 (63.0)	0.96
**Baseline blood glucose** (mmol/L), median (Q1, Q3)	10.8 (7.9, 14.0)	9.0 (7.7, 12.5)	0.50
**Hemoglobin** (g/L), median (Q1, Q3)	110.0 (88.0, 125.0)	109.5 (93.0, 124.0)	0.79
**INR**, median (Q1, Q3)	1.2 (1.0, 1.3)	1.2 (1.1, 1.3)	0.89
**Platelets** (10^9^/L), median (Q1, Q3)	177 (123, 264)	163 (117, 241)	0.90
**Bilirubin** (μmol/L), median (Q1, Q3)	13.2 (6.9, 25.3)	14.1 (8.7, 27.1)	0.45
**Creatinine** (µmol/L), median (Q1, Q3)	118 (81, 253)	77 (67, 96)	0.001
**C-reactive protein** (mg/liter), median (Q1, Q3)	131.0 (47.6, 160.0)	149.0 (108.0, 207.0)	0.07
**Serum lipid profile** (mmol/liter), median (Q1, Q3)			
Triglycerides	1.2 (0.9, 2.2)	1.2 (0.9, 1.8)	0.48
Cholesterol Non-HDL cholesterol	2.9 (2.5, 3.3) 2.2 (1.8, 3.1)	2.2 (1.8, 2.7) 1.7 (1.4, 2.1)	0.001 0.003
HDL cholesterol	0.6 (0.4, 0.8)	0.5 (0.4, 0.7)	0.33
LDL cholesterol	1.3 (0.9, 1.7)	0.9 (0.7, 1.2)	0.01
**Albumin** (g/L), median (Q1, Q3)	29 (28, 31)	27 (23, 320	0.15
**Pre-albumin** (g/L), median (Q1, Q3)	0.10 (0.07, 0.12)	0.12 (0.09, 0.14)	0.11
**Transferrin** (g/L), median (Q1, Q3)	1.3 (1.0, 1.3)	1.2 (1.0, 1.5)	0.67
**Hemoglobin A1C****(%)**, median (Q1, Q3)	6.0 (5.8, 7.3)	5.5 (5.4, 6.2)	0.04
**24-h urinary nitrogen excretion** (g/day), median (Q1, Q3)	4.8 (2.4, 7.0)	7.2 (5.1, 11.4)	0.007
**Nitrogen balance** (g/day)	−2.1 (−4.3, −0.1)	−6.9 (−10.1, −2.4)	0.003
**Minute ventilation** (L), median (Q1, Q3)	9.4 (8.7, 10.6)	10.3 (7.9, 11.5)	0.56
**Maximum temperature** (°C), median (Q1, Q3)	37.1 (36.5, 37.6)	37.1 (36.6, 37.7)	0.75

BMI: body mass index; APACHE II: Acute Physiology and Chronic Health Evaluation II; INR: International normalized ratio; SOFA: Sequential Organ Failure Assessment; PaO_2_: FIO_2_ ratio: the ratio of partial pressure of oxygen to the fraction of inspired oxygen; Non-HDL cholesterol: Total cholesterol-HDL, HDL: High density lipoproteins; LDL: Low density lipoproteins; The denominators for all percentages is the N for each column. Continuous variables are represented as median (quartile 1 and quartile 3). * FFAs normal range: 0.1 to 0.45 mmol/L (females); 0.1 to 0.6 mmol/L (males). Q1: first quartile, Q3: third quartile.

**Table 2 nutrients-11-00384-t002:** Daily caloric intake, protein intake, insulin and glucose data in patients with high and normal free fatty acids (FFAs) level.

Variable	High FFAs Level *n* = 23	Normal FFAs Level *n* = 47	*p*-Value
**Calculated caloric requirement** (kcal/day), median (Q1, Q3)	1729 (1371, 2031)	1909 (1588, 2119)	0.12
**Daily caloric intake for the intervention duration—All patients**			
No. of kilocalories, median (Q1, Q3)	1065.6 (785.0, 1496.1)	1071 (810.0, 1368.5)	0.73
Percent of requirement, median (Q1, Q3)	61.5 (49.3, 84.2)	55.8 (50.2, 74.9)	0.64
**Daily caloric intake for the intervention duration—Permissive underfeeding**			
*n* (%)	12 (52.2)	26 (55.3)	0.80
No. of kilocalories, median (Q1, Q3)	838.7 (622.4, 1028.1)	992.6 (792.1, 1109.7)	0.12
Percent of requirement, median (Q1, Q3)	53.7 (42.5, 60.3)	53.1 (48.0, 57.0)	0.94
**Daily caloric intake for the intervention duration—Standard feeding**			
*n* (%)	11 (47.8)	21 (44.7)	0.80
No. of kilocalories, median (Q1, Q3)	1496.1 (1136.1, 1708.3)	1355.4 (1042.4, 1626.6)	0.63
Percent of requirement, median (Q1, Q3)	84.2 (63.8, 91.1)	74.9 (55.7, 88.5)	0.61
**Caloric source for the intervention duration—All patients** (kcal /day), median (Q1, Q3)			
Enteral	1027.2 (709.3, 1393.5)	997.2 (732.1, 1323.4)	0.99
Propofol	17.3 (0.0, 102.6)	70.3 (19.1, 134.2)	0.06
Intravenous dextrose	6.1 (0.0, 22.3)	0.0 (0.0, 24.4)	0.76
Total parenteral nutrition	0 (0, 0)	0 (0, 0)	1.00
**Calculated protein requirement** (g/day), median (Q1, Q3)	72 (60, 85)	88 (72, 99)	0.03
**Daily protein intake for the intervention duration**, median (Q1, Q3)			
No. of grams	55.4 (44.2, 66.4)	61.9 (42.8, 77.1)	0.37
Percent of requirement	88.1 (60.6, 88.8)	76.5 (59.6, 87.7)	0.62
**Fat-to-carbohydrate ratio**, median (Q1, Q3)	1.3 (0.6, 1.7)	0.7 (0.6, 1.1)	0.06
**Duration of intervention** (days), median (Q1, Q3)	12 (6, 14)	11 (9, 14)	0.41
**Co-interventions during study period**			
Insulin			
Use, *n* (%)	19 (82.6)	22 (46.8)	0.004
Dose (units/day), median (Q1, Q3)	18.9 (3.8, 37.5)	0.0 (0.0, 14.9)	0.003
**Blood glucose** (mmol/liter), median (Q1, Q3)	9.5 (7.6, 11.9)	7.7 (6.5, 10.3)	0.05
**Enteral formulae on day 1 ^†^**, *n* (%)			
Disease-non-specific	8 (34.8)	27(57.5)	0.06
Disease-specific	15 (68.2)	20(42.6)	
**Propofol dose** (mcg/kg/min), median (Q1, Q3)	1.2 (0.0, 9.0)	5.2 (1.4, 11.2)	0.06
**Medications given during the ICU stay**, *n* (%)			
Beta blockers	10 (43.5)	23 (48.9)	0.67
Aspirin	11 (47.8)	8 (17.2)	0.007
Angiotensin-converting enzyme inhibitors	4 (17.4)	5 (10.6)	0.43
Angiotensin II receptor blockers	1 (4.4)	0 (0.0)	0.15
Statins	10 (43.5)	5 (10.6)	0.002

^†^ Disease-non-specific formula: Osmolite, Jevity, Promote, Ensure plus, Resource, Ensure, Resource plus, Jevity (1.2); Disease-specific formula: Glucerna, Nutric hepatic, Nepro, Pulmocare, Novasource Renal, Peptamen (1.0), Peptamen (1.2), Suplena, Oxepa. Q1: first quartile, Q3: third quartile. ICU: intensive care unit.

**Table 3 nutrients-11-00384-t003:** Outcomes of patients with high and normal free fatty acids (FFAs) level.

Outcomes	High FFAs Level *n* = 23	Normal FFAs Level *n* = 47	Crude *p* Value	Adjusted OR ** (95%CI)	*p* Value
**Death by 28 days**, *n* (%)	5 (21.7)	4 (8.5)	0.12	1.8 (0.29, 11.25)	0.52
**Death by 90 days**, *n* (%)	5 (21.7)	8 (17.0)	0.60	0.49 (0.09, 2.60)	0.40
**Death by 180 days**, *n* (%)	5 (21.7)	8 (17.4)	0.66	0.49 (0.09, 2.60)	0.40
**Death in the ICU**, *n* (%)	3 (13.6)	2 (4.3)	0.18	3.67 (0.56, 23.80)	0.17
**Death in the hospital**, *n* (%)	5 (22.7)	6 (13.3)	0.35	1.42 (0.31, 6.54)	0.65
**New renal replacement therapy**, *n* (%)	6 (27.3)	2 (4.7)	0.009	4.41 (0.56, 34.7)	0.16
**Healthcare-associated infections**, *n* (%)	12 (52.2)	27 (57.5)	0.68	0.62 (0.21, 1.86)	0.40
**Urinary tract infection**, *n* (%)	4 (17.4)	6 (12.8)	0.60	0.33 (0.03, 3.97)	0.38
**Ventilator associated pneumonia**, *n* (%)	7 (30.4)	19 (40.4)	0.42	0.63 (0.16, 2.46)	0.50
				**Parameter estimate (95% CI)**	
**ICU length of stay (days)**, median (Q1, Q3)	15 (11, 22)	17 (11, 24)	0.45	−0.80 (−6.42, 4.81)	0.76
**Hospital length of stay (days)**, median (Q1, Q3)	39 (23, 62)	46.5 (28.0, 97.0)	0.33	6.28 (−27.74, 40.3)	0.71
**Duration of mechanical ventilation (days)**, median (Q1, Q3)	10 (7, 18)	12 (7, 21)	0.57	3.69 (−2.05, 9.43)	0.20
**ICU-free days**, median (Q1, Q3)	74 (54, 79)	72 (55, 78)	0.73	5.73 (−9.08, 20.54)	0.44
**RRT-free days**, median (Q1, Q3)	14 (8, 14)	14 (14, 14)	0.01	−0.33 (−2.40, 1.73)	0.75
**MV-free days**, median (Q1, Q3)	12 (7, 21)	10 (7, 18)	0.51	4.70 (−11.00, 20.41)	0.55

ICU: intensive care unit, MV: mechanical ventilation, RRT: renal replacement therapy, Q1: first quartile, Q3: third quartile. ** Adjusted for age, gender, BMI, APACHE II, diabetes, triglycerides, LDL cholesterol, HDL cholesterol, non-HDL cholesterol and medical admissions (vs. non-medical admissions).

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
