# Peer review of "Free Fatty Acids’ Level and Nutrition in Critically Ill Patients and Association with Outcomes: A Prospective Sub-Study of PermiT Trial"

_nutrients, 2019, doi:10.3390/nu11020384_

Round 1
Reviewer 1 Report
this is a well written paper that is easy to understand. the findings relating FFA levels to glucose metabolism are not unexpected but are unique given the patient population studied.
The relation ship to propofol infusion (usually in 10% lipid) and the implications of this can be expanded upon.
Author Response
03.02.2019
Prof. Dr. Lluis Serra-Majem
Editor-in-Chief
Nutrients
Dear Prof. Serra-Majem
Thank you for considering " Free Fatty Acids and Nutrition in Critically Ill Patients and Association with Outcomes " for publication in Nutrients.
We also thank the reviewers for their additional valuable comments, which has further improved the manuscript. We have addressed all the comments and tried to provide line-by-line response to them. All the modifications in the manuscript were made in track changes. A clean copy is also attached.
Sincerely,
Yaseen Arabi, MD, FCCP, FCCM
Chairman, Intensive Care Department
Medical Director, Respiratory Services
Professor, College of Medicine
King Saud Bin Abdulaziz University for Health Sciences
King Abdulaziz Medical City
ICU 1425 PO Box 22490 Riyadh, 11426
Kingdom of Saudi Arabia
+966(11)801-1111 Ext. 18855/18877
arabi@ngha.med.sa
Reviewer 1
This is a well written paper that is easy to understand. the findings relating FFA levels to glucose metabolism are not unexpected but are unique given the patient population studied.
Reply
Thank you for your acknowledgement
Comment
The relationship to propofol infusion (usually in 10% lipid) and the implications of this can be expanded upon.
Reply
The following statement was added to the discussion;
“The slightly lower dose of propofol given to patients with high-FFAs level was likely be related to being older and more susceptible to sedation, therefore, these patients would normally receive smaller doses of propofol”.
Reviewer 2 Report
Congratulations on this interesting substudy which adds to our understanding of free fatty acids in critical illness. I have only a few minor points to make:
1. need to include some interpretation of the difference in nitrogen balance and urinary N losses between groups. The anabolic effect of the extra insulin might be a potential factor?
2. discussion of propofol use should acknowledge that the high-FFA group were much older, therefore would normally receive much smaller doses of propofol, this is likely to be a major reason for the difference in the propofol doses given to the two groups.
3. the whole article needs some minor language editing for clarity. eg line 184 and 244 use 'baseline glucose' or 'initial glucose' rather than 'inclusion glucose', if this is what you mean here.
Author Response
03.02.2019
Prof. Dr. Lluis Serra-Majem
Editor-in-Chief
Nutrients
Dear Prof. Serra-Majem
Thank you for considering " Free Fatty Acids and Nutrition in Critically Ill Patients and Association with Outcomes " for publication in Nutrients.
We also thank the reviewers for their additional valuable comments, which has further improved the manuscript. We have addressed all the comments and tried to provide line-by-line response to them. All the modifications in the manuscript were made in track changes. A clean copy is also attached.
Sincerely,
Yaseen Arabi, MD, FCCP, FCCM
Chairman, Intensive Care Department
Medical Director, Respiratory Services
Professor, College of Medicine
King Saud Bin Abdulaziz University for Health Sciences
King Abdulaziz Medical City
ICU 1425 PO Box 22490 Riyadh, 11426
Kingdom of Saudi Arabia
+966(11)801-1111 Ext. 18855/18877
arabi@ngha.med.sa
Reviewer 2
Comments and Suggestions for Authors
Congratulations on this interesting sub study which adds to our understanding of free fatty acids in critical illness. I have only a few minor points to make:
Comment 1
Need to include some interpretation of the difference in nitrogen balance and urinary N losses between groups. The anabolic effect of the extra insulin might be a potential factor?
Reply
We agree with your implication. As a result, the following sentence has been added to the discussion section;
“Interestingly, patients with high FFAs level had less 24-hour urinary nitrogen excretion and less negative nitrogen balance. This may be related to lower muscle mass in this older population and more frequent insulin therapy”.
Comment 2.
Discussion of propofol use should acknowledge that the high-FFA group were much older, therefore would normally receive much smaller doses of propofol, this is likely to be a major reason for the difference in the propofol doses given to the two groups.
Reply
Thank you for the suggestion. We have added the following statement to the discussion;
“The slightly lower dose of propofol given to patients with high-FFAs level was likely be related to being older and more susceptible to sedation, therefore, these patients would normally receive smaller doses of propofol”.
Comment 3
The whole article needs some minor language editing for clarity. eg line 184 and 244 use 'baseline glucose' or 'initial glucose' rather than 'inclusion glucose', if this is what you mean here.
Reply
Thank you for the suggestion. The article has been edited as suggested and the word “inclusion” has been replaced with “baseline” wherever application.